

# Synergy between TROPOMI sun-induced chlorophyll fluorescence and MODIS spectral reflectance for understanding the dynamics of gross primary productivity at integrated carbon observatory system (ICOS) ecosystem flux
**sites**

Hamadou Balde[1,2], Gabriel Hmimina[1], Yves Goulas[1], Gwendal Latouche[2], Kamel Soudani[2]

[1]Laboratoire de Météorologie Dynamique, Sorbonne Université, IPSL, CNRS/Ecole Polythétique, 91128, Palaiseau Cedex, France

[2]Ecologie Systématique et Evolution, Université Paris-Saclay, CNRS, AgroParisTech, 91190, Gif-sur-Yvette,
France

*Correspondence to*: Hamadou Balde (hamadou.balde@lmd.ispl.fr)

**Abstract:** An accurate estimation of vegetation Gross Primary Productivity (GPP), which is the amount of carbon taken up by vegetation through photosynthesis for a given time and area, is critical for understanding terrestrial-atmosphere $CO_2$ exchange processes, ecosystem functioning, and as well as ecosystem responses and adaptations
to climate change. Earliest studies, based on ground, airborne and satellite Sun-Induced chlorophyll Fluorescence (SIF) observations have recently revealed close relationships with GPP at different spatial and temporal scales and across different plant functional type (PFT). However, questions remain regarding whether there is a unique relationship between SIF and GPP across different sites and PFT and how can we improve GPP estimates using solely remotely sensed data. Using concurrent measurements of daily TROPOMI (TROPOspheric Monitoring
Instrument) SIF (daily $SIF_d$), daily MODIS Terra and Aqua spectral reflectance, and vegetation indices (VIs, notably NDVI (normalized difference vegetation index), NIRv (near-infrared reflectance of vegetation) and PRI (photochemical reflectance index)) and daily tower-based GPP across eight major different PFT, including mixed forests, deciduous broadleaf forests, croplands, evergreen broadleaf forests, evergreen needleleaf forests, grassland, open shrubland, and wetland, the strength of the linear relationships between tower-based GPP and $SIF_d$
at 40 ICOS (Integrated Carbon Observation Systems) flux sites was investigated, and the synergy between these variables to improve GPP estimates using a data-driven modelling approach was evaluated. The results revealed that the strength of the linear relationship between GPP and $SIF_d$ was strongly site-specific and PFT-dependent. Furthermore, the GLM (Generalized Linear Model) model, fitted between $SIF_d$, GPP, site and vegetation type as categorical variables, further supported this site-and PFT-dependent relationship between GPP and $SIF_d$. This study
also showed that the spectral reflectance bands (RF-R), $SIF_d$ plus spectral reflectance (RF-SIF-R) models explained over 80% of the seasonal and interannual variations in GPP, whereas the $SIF_d$ plus VIs (RF-SIF-VI) model reproduced only 75% of the tower-based GPP variance. In addition, the relative importance results demonstrated that the spectral reflectance bands in the near-infrared, red and $SIF_d$ appeared as the most influential and dominant factors determining GPP predictions, indicating the importance of canopy structure, biochemical properties and
vegetation functioning on GPP estimates. Overall, this study provides insights into understanding the strength of the relationships between GPP and SIF and the use of the spectral reflectance and $SIF_d$ to improve GPP across sites and PFT.



## 1. Introduction

In the context of climate change, understanding the role of terrestrial ecosystems in terms of exchanges of carbon, water and energy is crucial in order to fill-in the knowledge gap on climatic interactions between the biosphere and the atmosphere. Terrestrial ecosystems are one of the main components of the carbon cycle and are highly sensitive to abiotic stresses. Therefore, an accurate estimation of vegetation Gross Primary Productivity (GPP), which is the amount of flux carbon taken up by vegetation through photosynthesis, is critical for understanding terrestrial-atmosphere $CO_2$ exchange processes, ecosystem functioning, as well as ecosystem responses and adaptations to climate change (Gamon et al., 2019). Eddy Covariance (EC) techniques allow the estimation of GPP locally (Falge et al., 2002; Moureaux et al., 2008; Chu et al., 2021). However, they have limitations when it comes to upscale carbon fluxes estimates at larger scales due to their restricted spatial coverage, temporal dynamics of flux footprints, and limited distribution across different vegetation types, notably in key areas such as Africa and South America (X. Xiao, 2004; Gamon, 2015; J. Xiao et al., 2019). GPP can also be estimated based on physical and ecophysiological modelling approaches. However, for estimating GPP at larger scales, those methods are hampered by the lack of understanding of the underlying physiological processes (Jiang & Ryu, 2016; Y. Zhang et al., 2017; Madani et al., 2020).

Remote sensing is widely used to upscale canopy GPP to landscape, regional, and global scales and at daily scale using reflected sunlight measured by satellite sensors (Running et al., 2004; Baldocchi et al., 2020 ; Wu et al., 2020 ; Kong et al., 2022; X. Wang et al., 2022). These approaches are mainly based on reflectance-based indices (VIs) such as Normalized Difference Vegetation Index (NDVI), Enhanced Vegetation Index (EVI) and more recently near-infrared reflectance of vegetation (NIRv) (Baldocchi et al., 2020). However, VIs are mostly sensitive to spatial and temporal variability in structural (LAI (leaf area index), APAR (absorbed photosynthetically active radiation by chlorophyll), etc.) and biochemical canopy characteristics (Dechant et al., 2020; Pabon-Moreno et al., 2022). Although, they suffer from contamination by atmosphere and saturation in canopy dense ecosystems and are less sensitive to diurnal and daily variations in photosynthetic status resulting from physiological responses induced by rapid changes of abiotic stresses (Daumard et al., 2012; Guanter et al., 2014; Wieneke et al., 2016; Y. Zhang, Zhang, et al., 2021). Remote sensing also provides access to variables which are related to canopy functioning such as the photochemical reflectance index (PRI) (Gamon et al., 1992; X. Wang et al., 2020) and Sun-Induced chlorophyll Fluorescence (SIF) (Porcar-Castell et al., 2014; Goulas et al., 2017; Magney et al., 2019; P. Yang et al., 2020; J. Zhang et al., 2022; X. Li & Xiao, 2022).

PRI is a reflectance-based index, that has been shown to detect vegetation functioning activities under abiotic stress conditions that above-mentioned VIs cannot capture (Meroni et al., 2008). It is due to changes in the absorptance of leaves around 510 nm or reflectance at 531 nm that are related to the interconversion of the xanthophyll pigment cycles, which represents an important photoprotection mechanism (Gamon et al., 1992; Meroni et al., 2008). Moreover, previous studies pointed out that PRI can be used to improve canopy GPP estimates at the ecosystem level at daily timescale (X. Wang et al., 2020; Hmimina et al., 2015; Soudani et al., 2014), but how variations in PRI at long timescales with spatial variations of vegetation types affect the relationship between PRI and GPP remains unresolved and an area of active research (Porcar-Castell et al., 2014; Chou et al., 2017; Gitelson et al., 2017).





In recent years, SIF has emerged as a promising remotely sensed tool for monitoring canopy GPP, which is functionally and fundamentally different from the aforementioned VIs (Damm et al., 2010; X. Yang et al., 2015; Köhler et al., 2018; N. Wang et al., 2021; Guanter et al., 2021). In fact, SIF does not rely on vegetation reflectance, instead it is a faint signal directly emitted by chlorophyll from the absorbed sunlight just before the occurrence of

photochemical reaction (Porcar-Castell et al., 2014; Gu et al., 2019; Y. Zhang, Migliavacca, et al., 2021). SIF has a physical and physiological meanings, and hence SIF offers new opportunities for global assessment of canopy GPP (Mohammed et al., 2019; Wieneke et al., 2018; Z. Zhang et al., 2020; Kimm et al., 2021; Dechant et al., 2022). Early studies relied on ground-based, airborne and satellite SIF data measurements at different temporal and spatial scales have indicated a strong linear site-specific and vegetation types dependent relationship between

GPP and SIF (Frankenberg et al., 2011; Guanter et al., 2014; H. Yang et al., 2017; Wood et al., 2017; X. Li, Xiao, He, et al., 2018; Paul-Limoges et al., 2018; Y. Zhang, Zhang, et al., 2021; J. Zhang et al., 2022). In contrast, at finer temporal scales such as diurnal and hourly, the relationship between GPP and SIF is not as strong as at longer timescales. Instead, it appears to be non-linear due to rapid changes in instantaneous variations in PAR and environmental conditions (Damm et al., 2015; Marrs et al., 2020; Kim et al., 2021). How and at which extent

driving factors such as canopy structure, spatial heterogeneity and abiotic stress conditions mediate the GPP and SIF relationship remains a challenge and needs to be investigated (Smith et al., 2018; N. Wang et al., 2021; X. Li & Xiao, 2022). The main drawback relates to the use of SIF to predict GPP at regional and global scales lies on the difficulty in the weak SIF signals retrieval requiring averaging over large time and spatial scales, and thus hampers detecting fine-scale dynamics needed to explain underlying processes (Gamon et al., 2019; Köhler et al.,

2021).Yet, the TROPOspheric Monitoring Instrument (TROPOMI) sensor, which is on board Sentinel 5-Precursor, represents a novel for understanding SIF variations as well as an opportunity to fully evaluate the potential of SIF to improve GPP estimates at the ecosystem scale as it provides a quiet high temporal resolution at daily scale (Köhler et al., 2018). In addition, the future satellite mission FLEX (Fluorescence Explorer) will provide on a single platform SIF at an unprecedented spatial resolution (300m) together with visible reflectance in the green,

red and far red spectral windows (Drusch et al., 2017).

The surface spectral reflectance, VIs and SIF can be used altogether to better characterize highly spatiotemporal dynamics in vegetation canopy structure, canopy biochemical properties and vegetation functioning as a response to frequent changes in abiotic conditions at the site and ecosystem scales. However, to the best of our knowledge, an attempt to study the synergy between those variables have not been comprehensively addressed. Owing to most

likely that the relationships between structural and functional components are not linear, and have complex interactions over time and space (Hilker et al., 2007; Sippel et al., 2018; Yazbeck et al., 2021; Pabon-Moreno et al., 2022; Kong et al., 2022). Therefore, a series of observations of SIF, surface spectral reflectance and VIs at the site et ecosystem scales could give insights about how SIF is related to GPP, and whether SIF and spectral reflectance, and VIs would make better model parameters, and provide additional information on understanding

the dynamics of GPP at the ecosystem scale and beyond.

The overarching objective of this work is to study the potential of SIF, spectral reflectance and VIs (namely NDVI, NIRv, and PRI) to estimate canopy GPP at the site and the ecosystem scales, and the synergy between these predictive variables. Specifically, this study primarily intends to evaluate at daily timescale the strength of the linear relationships between SIF and GPP at 40 ICOS flux sites, including several vegetation functional types

(mixed forests, deciduous broadleaf forests, croplands, evergreen broadleaf forests, evergreen needleleaf forests,



grassland, open shrubland, and wetland), and ultimately to examine the synergy between SIF, spectral reflectance and VIs to improve canopy GPP estimates based on data-driven modelling approach.

## 2. Materials and Methods

In this current section, the site characteristics and Eddy Covariance (EC) flux data are presented. Then, the remote
sensing data (TROPOMI, MODIS Aqua and Terra, and Copernicus Land Cover classification) used in the study are described. At last, data analysis methods used in this study are presented.

### 2.1 Study Sites and flux tower in-situ data

EC flux data were obtained through the Integrated Carbon Observation System (ICOS) data portal release 2018 and 2021 (https://www.icos-cp.eu/data-services). We screened over 70 ecosystem ICOS sites relying on the
availability of GPP data for each site with simultaneous TROPOMI SIF observations in the period from February 2018 to December 2020, and maintained 40 sites for analyses. The study sites encompass from a latitude 5.27 °N to 67.75 °N, including a diversity of plant functional types (PFT) based on the IGBP vegetation types classification given by ICOS PI sites: Mixed Forests (MF, 2 sites), Croplands (CRO, 9 sites), Deciduous Broadleaf Forests (DBF, 6 sites), Evergreen Broadleaf Forest (EBF, 2 sites), Evergreen Needleleaf Forests (ENF, 13 sites), Grasslands
(GRA, 3 sites), Open Shrubland (OSH, 1 site, which is actually a young vineyard plantation), and Wetlands (WET, 4 sites). The PFT at each site was confirmed by photointerpretation of pictures found in ICOS data portal database and Google Earth. Detailed information and references of these sites are provided in Supplementary Materials in Tab S1. Figure 1 presents the location of these study sites, except for GF-Guy site. In the analyses, we used daily GPP values computed as the sum of the half-hourly values estimated from each site. GPP data previously gap filled
by ICOS PI representing for a full year, which was the case for instance at *CH-Dav, FR-Bil, IT-SR2,* and *SE-Deg*, are filtered out and were not used in the analyses.

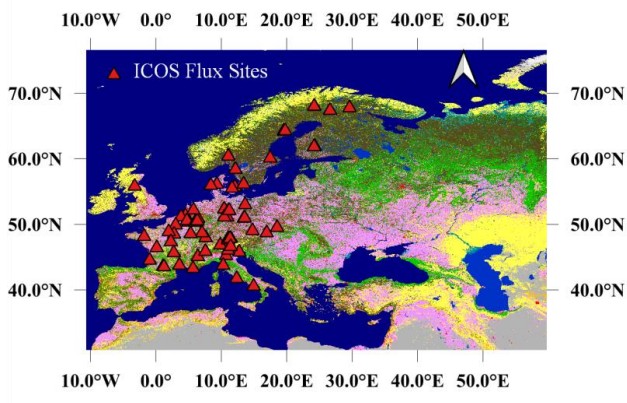

**Figure 1:** The study area and location of the EC ICOS flux sites, except for GF-Guy site. The base map is the 100 m spatial resolution of the Copernicus Global Land Cover Classification map. The triangles represent the locations of the flux sites used
for investigating the relationships between tower-based GPP and TROPOMI SIF.



### 2.2    Remote Sensing data

#### 2.2.1    MODIS Terra and Aqua Data

Timeseries of daily MODIS Terra and Aqua surface reflectance products (MOD09GA, MODOCGA, MYD09GA and MYDOCGA), centered at the location of each site, were downloaded from Google Earth Engine database. The

quality assurance (QA) flag (ideal quality, QA = 0) and the cloud mask (clear, cloud state = 0) criteria were used. Both MODIS Terra and Aqua contain 16 spectral bands of which, the spatial resolution from band 1 to band 7 is 500 m, and 1 km for the remaining bands (8-16) (Vermote et al., 2015). A detailed information about the MODIS data products is given in Supplementary Materials in Tab S2. We used daily MODIS surface reflectance, NDVI, NIRv, and PRI. These VIs are computed according the equation given in Table 1. For PRI computation, we used

$B_{13}$ as a reference band following (Hilker et al., 2009).

Table 1 : MODIS Terra and Aqua vegetation indices computations. $B_2$ (841-876 nm) denotes surface spectral reflectance at band 2, $B_1$ (620-670 nm) denotes surface spectral reflectance at band 1, $B_{11}$ (526-536 nm) represents the surface spectral reflectance at band 11, and $B_{13}$ (662-672 nm) represents the surface spectral reflectance at band 13.

| Acronym | Full Name | Formulation | Spatial Resolution | References |
|---|---|---|---|---|
| NDVI | Normalized Difference Vegetation Index | $(B_2 - B_1)/(B_2 + B_1)$ | 500 m | (Tucker, 1979) |
| PRI | Photochimal Reflectance Index | $(B_{11} - B_{13})/(B_{11} + B_{13})$ | 1 km | (Drolet et al., 2008; Hilker et al., 2009) |
| NIRv | Near-Infrared Reflectance of Vegetation | $B_2 \times NDVI$ | 500 m | (Badgley et al., 2017) |

#### 2.2.2    TROPOMI SIF and Copernicus Global Land Cover data

TROPOMI, as a single payload of the Sentinel-5 Precursor (S-5P) satellite, was launched on October 13, 2017. TROPOMI has a near sun-synchronous orbit with a repeat cycle of 16 days and an equatorial crossing time at around 13:30 local time (Köhler et al., 2018), which is comparable to those of OCO-2 (Orbiting Carbon Observatory-2) and GOSAT (Greenhouse Gases Observing Satellite). However, the wide swath of TROPOMI (2600 km) is larger than that of OCO-2 (10 km), which enables TROPOMI to provide almost daily spatially

continuous global coverage (Köhler et al., 2018). TROPOMI has a spatial resolution of 7 km along track (5 km since August 2019 owing to diminish integration time) and 3.5 to 14 km across track (based on the viewing angle) and covers the spectral range between 675-775 nm in the near infrared with a spectral resolution of 0.5 nm, which allows the retrieval of far-red SIF (Köhler et al., 2018). To decouple SIF emissions from the reflected incident sunlight, a statistical and data-driven approach is used, see Köhler et al. (2018) for more details. We used

instantaneous and daily ungridded soundings of TROPOMI far-red SIF at 740 nm obtained from Caltech dataset between February, 2018 and December, 2020 (https://data.caltech.edu/records/1347). Instantaneous SIF data were reported in (mW m$^{-2}$ sr$^{-1}$ nm$^{-1}$). Daily SIF (hereafter referred as SIF$_d$) is computed by timing instantaneous SIF with a day length correction factor included in the dataset.





The TROPOMI SIF observations corresponding to each site were determined relying on the following criteria.
Firstly, we extracted all pixels which center locations are less than 5 km away from the flux tower sites for analyses.
The latter choice was motivated due to the fact that the relationship between TROPOMI SIF and tower-based GPP
gradually weakened as the distance between sites to the center of pixels increased (data not shown). Secondly, to
reduce the cloud effects on SIF data, $SIF_d$ observations with cloud fraction over 15% were excluded, even though,
some findings reveal that TROPOMI SIF is less sensitive to cloud than surface reflectance values (Guanter et al.,
2012; Doughty et al., 2021). The 100 m spatial resolution of the Copernicus Global Land Cover Classification map
for the year 2019 (Buchhorn et al., 2020) was used as a based map of the study sites. This land cover classification
map was obtained from the Copernicus Global Land Service website (https://lcviewer.vito.be/download).

### 3. Data Analysis

In this study, the GPP and $SIF_d$ relationship was evaluated at the daily timescale at different spatial scales. Before
investigating the link between GPP and $SIF_d$, it was necessary to figure out a way to process outliers which were
mostly associated with negative $SIF_d$ values. It has been shown that excluding directly negative SIF values could
have effects on studying the relationships between satellite SIF data and GPP (Köhler et al., 2018; Köhler et al.,
2021). Thus, to handle the outliers, an exponential model was used to account for the structural relationship
between the instantaneous SIF and the SIF error included in the dataset. A threshold of $\pm 0.15$ mW m$^{-2}$ sr$^{-1}$ nm$^{-1}$ was
then applied to the residual random error of the exponential model.

The performance of $SIF_d$ to predict GPP using linear regression model at each site was examined. Afterward, a
PFT-specific $SIF_d$-GPP linear model was fitted to investigate the effects of PFT (plant functional type). To explore
the genericity of the linear relationship between GPP and $SIF_d$, first a linear regression model on data pooled across
all sites was adjusted. Second, to test further how the year, site and PFT, as categorical variables, and their
interactions (year*GPP, site*GPP, and PFT*GPP) influence the GPP and $SIF_d$ relationship, a Generalized Linear
model (GLM) was used. Within the GLM model, $SIF_d$ is considered as a response variable, whereas, site, PFT,
year and GPP are the explanatory variables. These aforementioned variables and their interaction effects may affect
the changes or variations either in $SIF_d$ or GPP and consequently influence the slope and intercept of their
relationships.

In order to study the synergy between $SIF_d$, spectral reflectance and VIs to improve GPP estimates, a Random
Forest (RF) regressor ensemble decision tree model was used (Brieman, 2001). Briefly, a RF is a machine learning
algorithm, which combines the results of several randomly ensemble decision trees to reach a final accurate output.
Before setting up the RF model, the correlation matrix between all variables was computed. It has been shown that
features importance can be affected by the high correlation between feature predictors (Toloşi & Lengauer, 2011),
suggesting that a decrease in importance values is observed when the level of correlation and the number of
correlated variables increases. In practice, a strongly predictive variable belonging to a group of correlated
variables can be considered less important than an independent and less informative variable. Based on remotely
sensed data inputs and one categorical explanatory variable (PFT), what variables are the most relevant for
estimating GPP on daily data pooled altogether across all sites were evaluated. Four RF models were established
relying on the combination of the predictive variables to estimate GPP: (1) only surface spectral reflectance (RF-
R), (2) surface spectral reflectance plus $SIF_d$ (RF-SIF-R), (3) surface spectral reflectance plus $SIF_d$ and the PFT as
categorical variable (RF-SIF-R-PFT), and (4) $SIF_d$ plus VIs (RF-SIF-VI) (namely NDVI, NIRv, and PRI). 80 per



cent of the data were used for training and the remaining for testing the model. It is worth mentioning that a RandomizeSearchCV technique was used (sklearn.model_selection.RandomizedSearchCV — scikit-learn 1.1.1 documentation) to tune the model and took the best parameters for each model to predict GPP and applied a 10-fold cross-validation and 20 iterations on the training set to avoid splitting the dataset into training, validating and testing sets which could affect the amount of data allocated for the training and could lead easily to model overfitting. The ensemble of decision tree models includes 200 trees for all models, but the number of splits per tree and the maximum depth varied. The relative importance of each variable, based on the mean decrease in impurity method, was used to evaluate the part of the contribution of each input variable in predicting the canopy GPP variability. For TROPOMI data extraction, MATLAB R2021a (The MathWorks, Inc., USA) was used and python version 3.9.1 was used for data analysis and visualization (sklearn, scipy, seaborn, matplotlib, pandas, and numpy).

Ultimately, the coefficient of determination ($R^2$), Root Mean Squared Error (RMSE), and the p-value metrics were used to evaluate the power of the linear agreement between GPP and $SIF_d$ for the site- specific and PFT-specific relationships. The aforementioned metrics plus the adjusted coefficient of determination (adj. $R^2$) were also used to evaluate the performance of the different RF models between the observed and predicted GPP. At last, but not least, a paired t-test is used to compare the performance of the RF models based on the method proposed by (Nadeau et al., 2003). A 5% significance level was used for all statistical inference.

## 4.    Results

### 4.1    GPP vs $SIF_d$ relationships

#### 4.1.1    Site-specific relationships

The first aim was to evaluate the strength of the linear relationship between tower-based GPP and $SIF_d$ encompassing different vegetation types at site level. Figure 2 shows the relationship between GPP and $SIF_d$ at each site. Overall, $SIF_d$ was significantly related with tower-based GPP at the site level and at the daily timescale (as $p<0.0001$ was statistically highly significant), except for *IT-Cp2* site of which GPP and $SIF_d$ relationship was insignificant and weak ($R^2 = 0.001$, $p≤0.60$). Furthermore, the Figure 2 indicates that the slopes and intercepts of the linear regression between GPP and $SIF_d$ are site-independent, suggesting that the difference in plant functional types and spatial heterogeneity across sites may significantly affect the relationship between GPP and $SIF_d$. The strongest relationships were found at *DK-Sor* and *FR-Fon*, which are DBF vegetation type sites, with $R^2$ values of 0.81 ($p<0.0001$) and 0.66 ($p<0.0001$), respectively, while the lowest linear relationships were recorded in EBF and CRO, namely at *GF-Guy*, *IT-Cp2* and *FR-Mej* sites, with $R^2$ being 0.02, 0.001 and 0.04, respectively. For each fit, the numbers of data points were between 178 and 1594, depending on the data availability at each site. A detailed information and statistics on the linear relationships between GPP and $SIF_d$ at each site is given in Supplementary Materials in Tab S3.





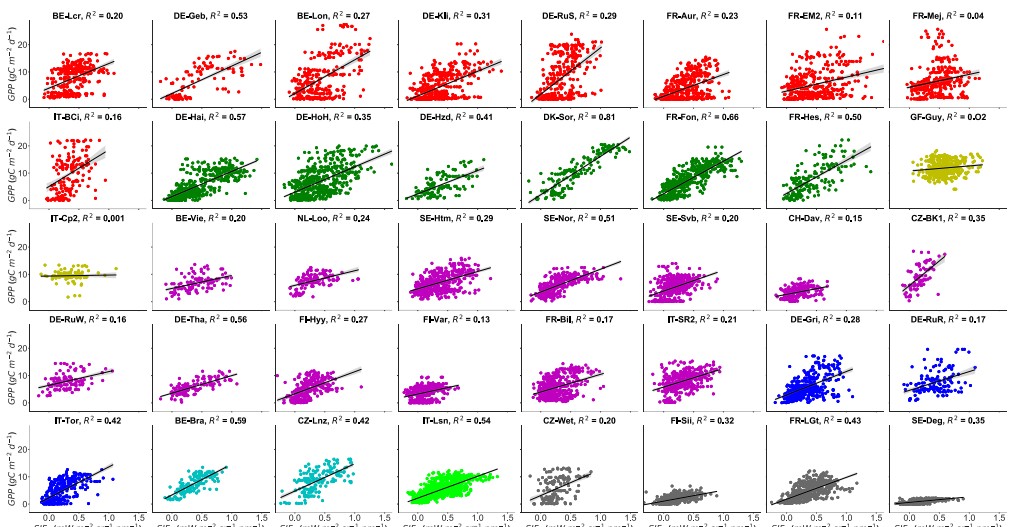

**Figure 2:** Site-specific tower-based GPP and $SIF_d$ relationships at daily timescale. The $R^2$ represents the coefficient of determination of the relationship between GPP and $SIF_d$ for each site. All pairwise linear relationships between GPP vs $SIF_d$ were statistically significant with p<0.0001. The color code represents the eight different plant functional types encountered in the study sites: Red color stands for CRO (croplands), green for DBF (deciduous broadleaf forests), yellow for EBF (evergreen broadleaf forests), magenta for ENF (evergreen needleleaf forests), blue for GRA (grasslands), Cyan for MF (mixed forests), lime for OSH (open shrubland), and dimgrey for WET (wetland). The shaded area depicted in each line is the 95% confidence interval of the linear relationship between GPP and $SIF_d$.

### 4.1.2 Plant functional type-specific and overall sites relationships

To test the effects of the PFT on the relationship between GPP and $SIF_d$ at the daily timescale, data were pooled across sites of the same PFT (MF, CRO, ENF, DBF, EBF, GRA, OSH, and WET) and a linear regression model was applied on each PFT. Figure 3 depicts the scatterplots of the relationships between GPP and $SIF_d$. The linear relationship between GPP and $SIF_d$ was statistically significant for all PFT ($R^2$ = 0.07-0.54, p<0.0001), taken individually. Furthermore, the slope of the linear regression between GPP and $SIF_d$ was strongest for DBF and MF (10.75±0.33 and 10.53±0.87 gC m$^{-2}$ d$^{-1}$/ (mW m$^{-2}$ sr$^{-1}$ nm$^{-1}$)), respectively) and the lowest for EBF (3.08±0.72 gC m$^{-2}$ d$^{-1}$/ (mW m$^{-2}$ sr$^{-1}$ nm$^{-1}$)). It can also be seen from the figures that the slopes and the intercepts of their linear relationships were clearly PFT-specific, as shown in Table 2.

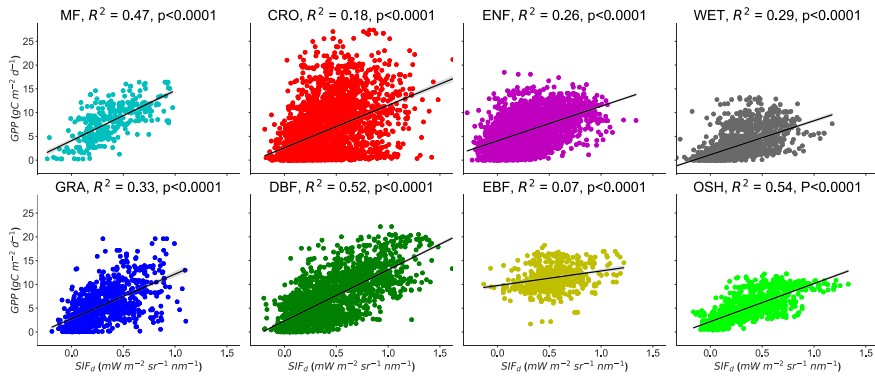



**Figure 3:** Linear relationships between tower-based GPP and $SIF_d$ in eight plant functional types: MF, CRO, ENF, DBF, EBF, GRA, OSH, and WET at daily timescale. The $R^2$ represents the coefficient of determination of the relationship between GPP and $SIF_d$. p is the probability value of the linear model. The shaded area depicted in each line is the 95% confidence interval of the linear relationships between GPP and $SIF_d$.

**Table 2:** Summary statistics of plant functional type-specific GPP and $SIF_d$ relationship in eight major PFT. All pairwise linear relationships between GPP and $SIF_d$ were statistically significant with $p < 0.0001$. The units are for the slope in (gC m$^{-2}$ d$^{-1}$/ (mW m$^{-2}$ sr$^{-1}$ nm$^{-1}$)), intercept in (gC m$^{-2}$ d$^{-1}$), and RMSE in (gC m$^{-2}$ d$^{-1}$). The sign $\pm$ denotes the confidence interval on the slope and intercept of the relationships between $SIF_d$ and GPP.

| PFT | Sites | $R^2$ | Slope | Intercept | RMSE | N |
|---|---|---|---|---|---|---|
| CRO | 9 | 0.18 | 8.93±0.49 | 2.61±0.24 | 5.24 | 5726 |
| DBF | 6 | 0.52 | 10.75±0.33 | 2.32±0.18 | 3.50 | 3762 |
| EBF | 2 | 0.07 | 3.08±0.72 | 9.76±0.42 | 2.62 | 964 |
| ENF | 13 | 0.26 | 7.28±0.28 | 4.07±0.10 | 2.85 | 7066 |
| GRA | 3 | 0.33 | 9.39±0.62 | 2.79±0.24 | 3.24 | 1768 |
| MF | 2 | 0.47 | 10.53±0.87 | 4.10±0.39 | 2.74 | 646 |
| OSH | 1 | 0.54 | 8.00±0.36 | 2.17±0.17 | 2.10 | 1594 |
| WET | 4 | 0.29 | 7.14±0.40 | 1.15±0.14 | 2.44 | 2950 |
| ALL | 40 | 0.30 | 9.12±0.17 | 2.87±0.08 | 3.82 | 24476 |

Moreover, the genericity of the linear relationship between GPP and $SIF_d$ was first tested on data pooled together across all sites (Figure 4). A significant but weak relationship between GPP and $SIF_d$ was found across all sites with $R^2$ of 0.30 ($p < 0.0001$) and RMSE of 3.82 gC m$^{-2}$ d$^{-1}$. However, when the variations between the year, site and PFT as inputs variables were included in a GLM model, along with GPP, the results showed a strong significant relationship between $SIF_d$, year, site, PFT and GPP ($p < 0.001$). Furthermore, the interactions between year and GPP, PFT and GPP were found to have statistically substantial effect on $SIF_d$ and GPP relationship, while the interaction between site and GPP was not significant (see Supplementary Material in Tab S4). These findings indicate that the slope of the GPP and $SIF_d$ relationship is considerably influenced by the site PFT and the interannual variations of $SIF_d$.

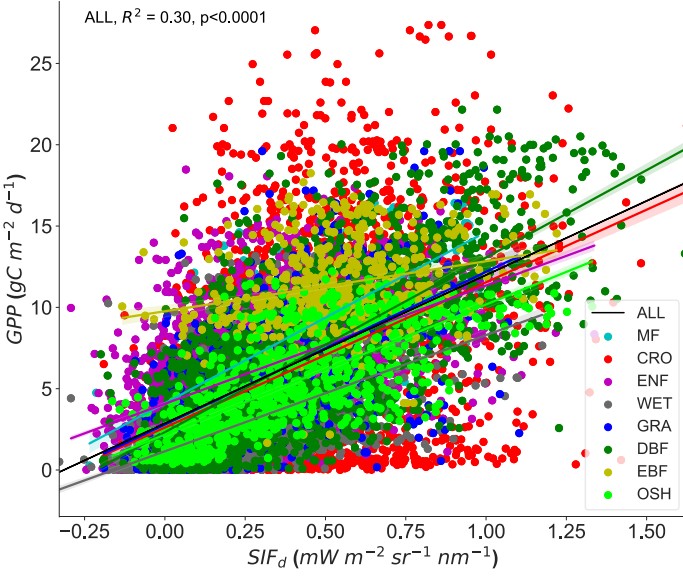



**Figure 4:** Scatterplots of the linear relationships between tower-based GPP and $SIF_d$ in eight PFT pooled together across all sites. The shaded area depicted in each line is the 95% confidence interval of the linear relationships between GPP and $SIF_d$.

### 4.2    Synergy between SIF, surface spectral reflectance and reflectance-based indices to quantify GPP

In order to optimise the inputs for the Random Forest (RF) regression and avoid the effects of high correlated explanatory variables on the model performance, the correlation matrix was computed. The correlation matrix (supplied in Supplementary Materials Figure S1) revealed a strong dependency between predictive variables (notably $B_9$ vs $B_{10}$, $B_{11}$ vs $B_{12}$ and $B_{13}$ vs $B_{14}$), indicating that using a RF model built in these variables could be affected by those high correlations. Based on these observations, the spectral reflectance of $B_{10}$, $B_{12}$ and $B_{14}$ were excluded from the explanatory variables of RF regression models.

### 4.2.1    Performance of GPP estimates using Random Forest regression

In Figure 5, it is represented tower-based GPP against the four RF GPP models across all sites. Overall, all the RF models predicted GPP show a high agreement with tower-based GPP. Yet, the RF-R model has the strongest relationship with tower-based GPP with an adjusted $R^2$ of 0.86 and RMSE of 1.72 gC $m^{-2}$ $d^{-1}$, while the RF-SIF-VI model presents the lowest predictions of GPP as the adjusted $R^2$ and RMSE were 0.75 and 2.29 gC $m^{-2}$ $d^{-1}$, respectively. Furthermore, the RF-SIF-R and RF-SIF-R-PFT model performed similarly well on estimating GPP as they could explain 82% and 83% of the variations in GPP across all sites, respectively. A paired t-test realized between the four models based on the adjusted $R^2$ performance revealed that the difference in mean between RF-R and RF-SIF-R, RF-R and RF-SIF-R-PFT, and RF-SIF-R and RF-SIF-R-PFT models was not statistically significant. In other words, these three FR models have the same performance.

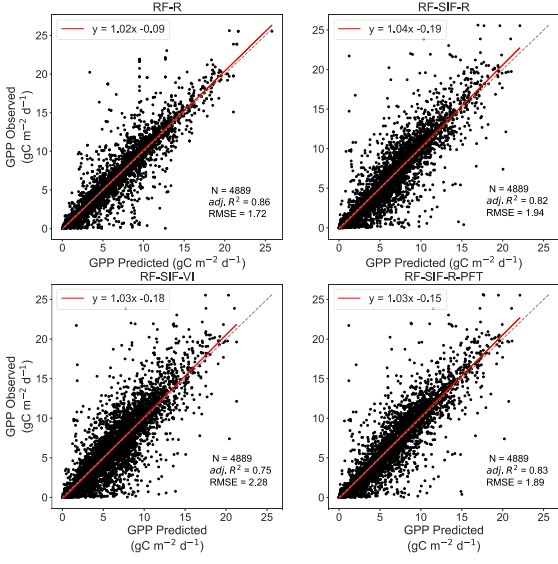



**Figure 5:** scatterplots of the observed GPP against the RF predicted GPP across all sites. The N denotes the number of data points used for the RF model's testing, adj. $R^2$ represents the adjusted coefficient of determination of the linear relationship between observed GPP and predicted GPP, and the RMSE is the Root Mean Squared Error between observed GPP and RF model predicted GPP. The dash diagonal line depicts the 1:1 line. RF-R denotes GPP prediction using only surface spectral reflectance, RF-SIF-R includes spectral reflectance and $SIF_d$ as inputs to predict GPP, RF-SIF-VI

integrates $SIF_d$ and VIs to estimate GPP, and RF-SIF-R-PFT includes surface spectral reflectance, $SIF_d$ and plant functional type as categorical variable to predict GPP.

The RF regression model's GPP estimates and the observed GPP representing different vegetation types at the site level are depicted in the Figures 6 and 7 for the RF-SIF-R model predictions as an example. The estimates for each site from the others models are presented in the Supplementary Materials (Figures S3-a RF-R, S3-b

RF-R, S4-a RF-SIF-VI, S4-b RF-SIF-VI, S5-a RF-SIF-R-PFT and S5-b RF-SIF-R-PFT) and the summary statistics results in Tab S5 for all RF models. At the site level, the RF-SIF-R model predicted tower-based GPP with high accuracy ($R^2$ = 0.55-0.95), except for three sites such as *IT-BCi* ($R^2$ = 0.22), *IT-Cp2* ($R^2$ = 0.27), and *SE-Deg* ($R^2$ = 0.41), where the RF-SIF-R model has difficulties to reproduce GPP. It is worth noting that all others RF models have a poor GPP predictions for these aforementioned sites. However, on data pooled

across all sites of the same PFT, the RF-SIF-R model show high performance in estimating GPP for all eight major PFT with $R^2$ being between 0.68 and 0.90. The lowest predictions are encountered in CRO and EBF sites, whereas the best tower-based GPP estimates were found in DBF and OSH sites.

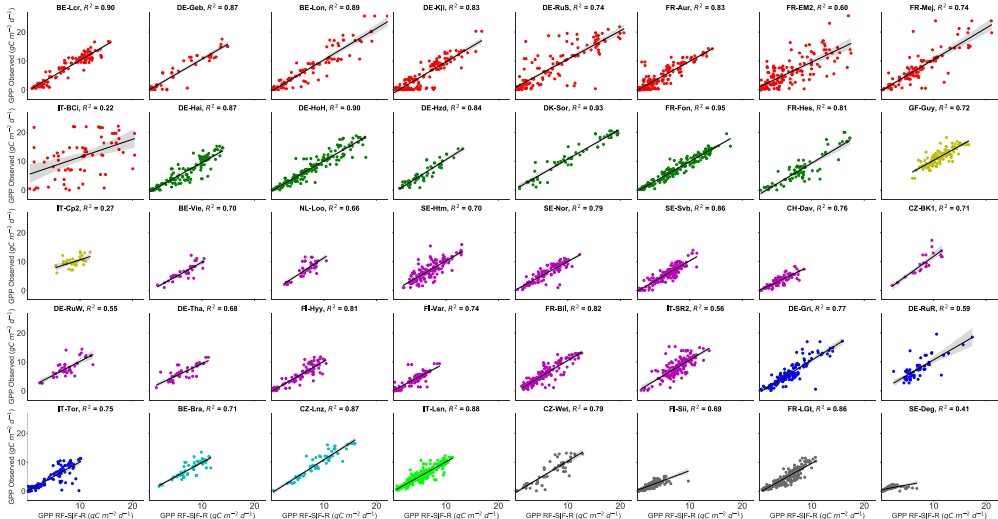

**Figure 6:** Site-specific scatterplots between observed GPP and RF-SIF-R predicted GPP at daily timescale. The $R^2$ represents the coefficient of determination of the linear relationship between observed GPP and predicted GPP. All pairwise linear relationships between observed GPP vs predicted GPP were statistically significant at all sites (with p<0.0001). The color code represents the eight different vegetation types encountered in the study sites: Red color stands for CRO, green for DBF, yellow for EBF, magenta for ENF, blue for GRA, Cyan for MF, lime for OSH, and dimgrey for WET.





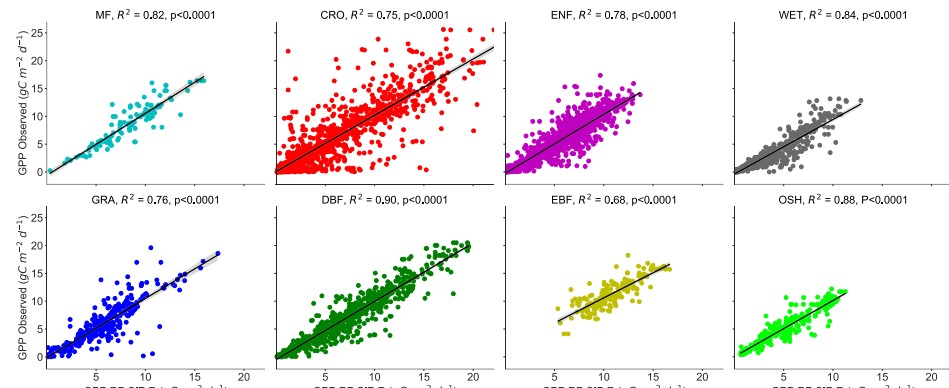

**Figure 7:** Scatterplots of observed GPP against RF-SIF-R predicted GPP in eight PFT at daily timescale. The $R^2$ represents the coefficient of determination of the linear relationship between observed GPP and predicted GPP. p denotes probability value of the linear relationships.

In Figure 8 and Table 3, it is depicted the observed and estimated GPP representing different PFT for all four RF models. The estimation for each site is given in Supplementary Materials Figure S2. Overall, all RF models' GPP predictions capture very well the seasonal and interannual dynamics of the tower-based GPP. However, there are sites, years and vegetation types where observed GPP cannot be estimated with high accuracy. For instance, the RF models tend to underestimate GPP maxima in GRA, WET and EBF vegetation types. These underestimates are mostly marked by the slope of the relationships between the observed GPP and predicted GPP in Table 3.

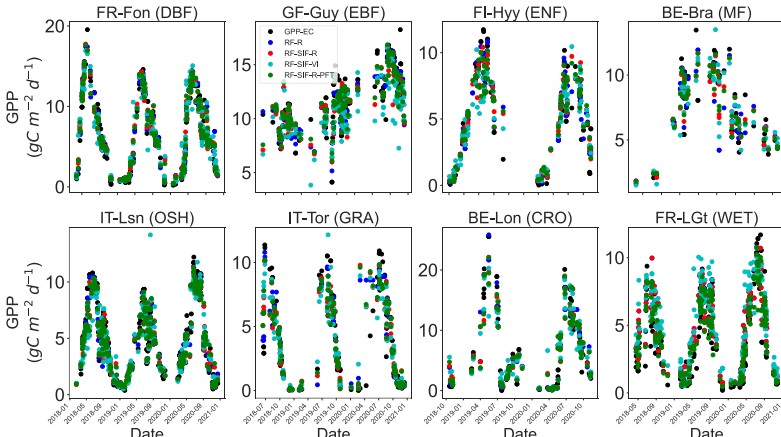

**Figure 8:** Comparison between observed GPP and RF regression models estimated GPP at selected ICOS flux sites representing different PFT: DBF, EBF, ENF, MF, CRO, GRA, OSH, and WET. The color code represents the different RF GPP predictions and the observed GPP: Red color stands for RF-SIF-R, green for RF-SIF-R-PFT, blue for RF-R, Cyan for RF-SIF-VI, and black for observed GPP.

**Table 3**: Summary statistics of plant functional type-specific observed GPP against RF models predicted GPP relationship in eight major PFT: MF, CRO, ENF, DBF, EBF, GRA, OSH, and WET. All pairwise linear relationship between observed GPP and predicted GPP were statistically significant with p<0.0001. the sign ± denotes the confidence interval on the slope and intercept of the relationships between observed GPP and predicted GPP.





| | | RF-R | | | | RF-SIF-R | | | |
|---|---|---|---|---|---|---|---|---|---|
| PFT | Sites | R$^2$ | Slope | Intercept | RMSE | R$^2$ | Slope | Intercept | RMSE | N |
| CRO | 9 | 0.79 | 1.03±0.03 | 0.00±0.24 | 2.67 | 0.75 | 1.01±0.03 | 0.08±0.26 | 2.89 | 1171 |
| DBF | 6 | 0.92 | 1.02±0.02 | -0.23±0.18 | 1.41 | 0.90 | 1.05±0.02 | -0.52±0.21 | 1.61 | 748 |
| EBF | 2 | 0.77 | 0.93±0.07 | 1.01±0.83 | 1.23 | 0.68 | 0.90±0.09 | 1.58±0.99 | 1.45 | 188 |
| ENF | 13 | 0.85 | 1.01±0.02 | -0.01±15 | 1.29 | 0.78 | 1.06±0.03 | -0.23±0.19 | 1.54 | 1385 |
| GRA | 3 | 0.82 | 1.02±0.05 | -0.02±32 | 1.64 | 0.76 | 1.07±0.06 | -0.17±0.38 | 1.87 | 364 |
| MF | 2 | 0.84 | 1.05±0.08 | -0.15±0.76 | 1.49 | 0.82 | 1.12±0.10 | -0.62±0.83 | 1.56 | 117 |
| OSH | 1 | 0.91 | 1.02±0.04 | -0.09±0.22 | 0.99 | 0.88 | 1.01±0.04 | 0.01±0.24 | 1.10 | 317 |
| WET | 4 | 0.92 | 0.98±0.02 | -0.15±0.10 | 0.85 | 0.84 | 0.98±0.03 | -0.37±0.15 | 1.17 | 599 |
| ALL | 40 | 0.86 | 1.02±0.01 | -0.09±0.08 | 1.72 | 0.82 | 1.04±0.01 | -0.19±0.10 | 1.94 | 4889 |
| | | RF-SIF-VI | | | | RF-SIF-R-PFT | | | |
| PFT | Sites | R$^2$ | Slope | Intercept | RMSE | R$^2$ | Slope | Intercept | RMSE | N |
| CRO | 9 | 0.70 | 1.03±0.04 | 0.01±0.29 | 3.14 | 0.75 | 1.00±0.03 | 0.12±0.26 | 2.87 | 1171 |
| DBF | 6 | 0.84 | 1.05±0.03 | -0.58±0.28 | 2.06 | 0.91 | 1.04±0.02 | -0.40±0.21 | 1.56 | 748 |
| EBF | 2 | 0.51 | 0.77±0.11 | 3.42±1.14 | 1.80 | 0.72 | 0.96±0.09 | 0.74±0.98 | 1.37 | 188 |
| ENF | 13 | 0.66 | 1.02±0.04 | 0.10±0.24 | 1.92 | 0.79 | 1.08±0.03 | -0.39±0.19 | 1.5 | 1385 |
| GRA | 3 | 0.70 | 0.98±0.07 | 0.02±0.43 | 2.11 | 0.77 | 1.07±0.06 | -0.29±0.38 | 1.84 | 364 |
| MF | 2 | 0.71 | 1.04±0.12 | 0.04±1.07 | 2.00 | 0.83 | 1.12±0.09 | -0.73±0.84 | 1.56 | 117 |
| OSH | 1 | 0.83 | 0.98±0.05 | 0.21±0.29 | 1.33 | 0.89 | 1.02±0.04 | -0.06±0.24 | 1.08 | 317 |
| WET | 4 | 0.72 | 0.88±0.04 | -0.39±0.21 | 1.54 | 0.88 | 1.05±0.03 | -0.29±0.12 | 0.99 | 599 |
| ALL | 40 | 0.75 | 1.03±0.02 | -0.18±0.12 | 2.28 | 0.83 | 1.03±0.01 | -0.15±0.09 | 1.89 | 4889 |

#### 4.2.2 Relative importance of the predictive variables on predicting GPP

Figure 9 shows the relative importance (or mean decrease in impurity) of the predictive variables of the RF models for predicting GPP across all sites pooled together. The Figure 9 indicates that for RF-R model, the surface spectral reflectance in the near-infrared (NIR) band (B$_2$ :841-876 nm) and the surface reflectance in the red band (B$_1$: 620-670 nm) were found as the most important inputs variables for GPP estimates. Moreover, it can be seen that the contribution of the far-red spectral reflectance (B$_{13}$) on predicting GPP is also important, whereas the contribution

of the others spectral reflectance bands was on similar level. For the RF-SIF-R model, SIF$_d$ (>23%), surface reflectance in the NIR (B$_2$ = 17%) and the surface spectral reflectance in the red band (B$_1$= 9%) are far largely the most relevant variables for GPP prediction, while the other variables contribute less into GPP estimates. The RF-SIF-R-PFT model differs with the previous model (RF-SIF-R) only on the plant functional type categorical variable and its results underline that the plant functional type variable is still important for predicting GPP. Ultimately,

reflectance-based vegetation indices are widely used for predicting GPP at larger scales. Hence, it is worthwhile investigating what are the contributions of these interesting variables jointly with SIF$_d$ in predicting canopy GPP. The relative importance derived from the RF-SIF-VI model reveals that SIF$_d$ (36%) is substantially the most relevant variable for predicting GPP. The contributions of NIR$_v$ and NDVI to the model are comparable, whereas PRI has a lower contribution in estimating GPP.



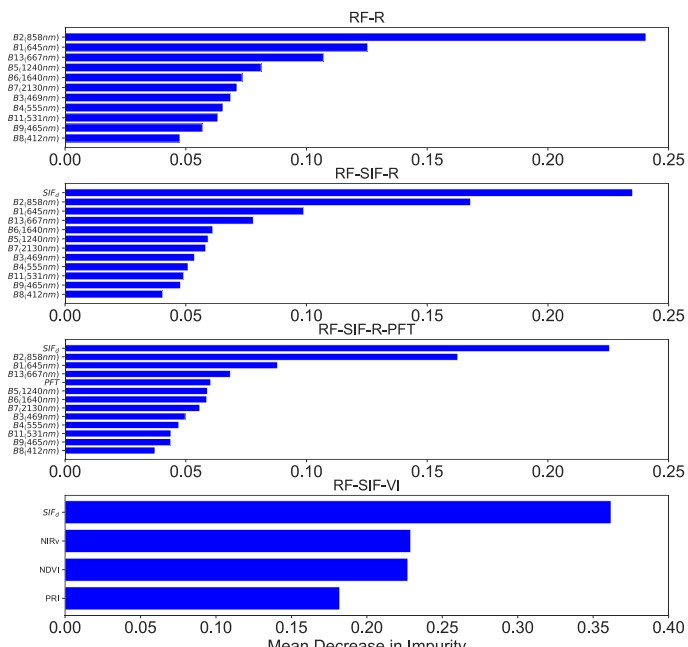

**Figure 9:** Relative importance of predictive variables of the RF models based only on remote sensing data for estimating GPP, except for the RF-SIF-R-PFT model. RF-R model based only on MODIS surface spectral reflectance bands, RF-SIF-R model uses SIF$_d$ and surface reflectance as input variables, RF-SIF-R-PFT model integrates SIF$_d$, surface reflectance and PFT as explanatory variables, and RF-SIF-VI model combines SIF$_d$ and reflectance-based indices notably NDVI, NIR$_v$, and PRI as input variables for predicting GPP across all sites. The wavelengths depicted on the spectral bands denote the central wavelength.

## 5. Discussions

### 5.1 Strength of the linear relationship between GPP and SIF$_d$ at site level and PFT scale

In this study, the first aim was to evaluate the strength of the relationship between tower-based GPP and SIF$_d$ in different PFT at daily timescale and different spatial scales (at site and vegetation type levels).

At the site level, the results demonstrate that there were strong and statistically significant relationships between GPP and SIF$_d$. However, the linear relation between tower-based GPP and SIF$_d$ across diverse sites vary significantly in terms of the slope and the intercept, which suggests a site-specific relationship. In other words, at these scales the differential variations in plant physiology and vegetation structure across sites and years and the spatiotemporal dynamics of the flux tower footprints (depending mainly on the height of the tower and wind direction), along with spatial heterogeneity and environmental conditions across sites may strongly affect first of all the SIF emissions, scattering and reabsorption across sites, and consequently the relationship between GPP and SIF$_d$ (Fournier et al., 2012; Paul-Limoges et al., 2018; Tagliabue et al., 2019; J. Li et al., 2020; Chu et al., 2021; Y. Zhang, Zhang, et al., 2021). These results are consistent with previous studies based on ground-based and satellite measurements which found evidence that canopy structure, as well as PFT have substantially great effects





on the relationship between GPP and $SIF_d$ across multiple sites (Dechant et al., 2020; Lu et al., 2020; X. Li, Xiao, He, et al., 2018; Sun et al., 2018; X. Wang et al., 2020; Hao et al., 2021; X. Wang et al., 2022). For instance, Wang et al. (2020) found that the relationship between OCO-2 SIF observed at 757 nm and 771 nm and tower-based GPP

across eight vegetation types at 61 flux sites all over the world relies on canopy structure and Lu et al. (2020) reported a better relationship between canopy GPP and SIF corrected from reabsorption and scattering effects than top of canopy SIF based on ground-based measurements, underlying the importance of canopy structure on SIF and GPP relationships.

The relationship between tower-based GPP and $SIF_d$ considering the PFT was also examined. The results revealed

a strong significant PFT-specific GPP and $SIF_d$ relationships across all eight major vegetation type. Yet, the slopes and the intercepts of GPP vs $SIF_d$ relationship were PFT-dependent. The slope of the relationship between GPP and $SIF_d$ is driven by the ratio of canopy photosynthesis and fluorescence yield, along with the canopy escape probability fraction of SIF photons from canopy to sensor (Porcar-Castell et al., 2014; Y. Zhang et al., 2018; Zeng et al., 2019). The major drivers affecting the canopy photosynthesis and SIF yield include among others leaf

morphology and orientation, plant physiology, canopy structure (leaf area index, chlorophyll contents, etc.), rapid changes in incident radiation and illuminated canopy surface, different contributions from photosystem I and II, as well as rapid abiotic responses (Porcar-Castell et al., 2014; Mohammed et al., 2019; Gamon et al., 2019; P. Yang et al., 2020; Chu et al., 2021; X. Wang et al., 2022). These explanations altogether sustained the PFT-specific GPP vs SIF relationship as those factors can considerably differ across PFT. Additionally, The results showed that the

DBF and OSH sites have the strongest GPP and $SIF_d$ relationship, which indicates that SIF may easily capture the seasonal, interannual and phenological variations in GPP within this vegetation type. On the other hand, the lower observed relations between GPP and $SIF_d$ in EBF (*GF-Guy & IT-Cp2*) sites could be partly explained by a lower spatiotemporal variability of SIF emissions in tropical forests coupled to a dispersed and lower GPP values observed on the datasets, as well as challenges in detecting or decoupling the understory vegetation effects from

all vegetation canopy contribution to SIF emissions and uncertainties related to GPP estimates in tropical forests, while in CRO (*FR-Mej & IT-BCi*) the difference in photosynthetic pathways ($C_3$, $C_4$ or mixed of both) and different management practices may be the reason why $SIF_d$ could not capture the variations in GPP, as reported in early studies (X. Li, Xiao, He, et al., 2018; Hayek et al., 2018; Mengistu et al., 2020; He et al., 2020; Hornero et al., 2021; X. Li & Xiao, 2022). Previous researches have also reported weak relationship between GPP and SIF in

EBF stands biome (X. Li, Xiao, & He, 2018; X. Wang et al., 2020). Moreover, it is worth mentioning that the biases related to cloudless sky and cloudy sky in space-based SIF retrieval, complicates the use of SIF to estimate GPP at the PFT scale because cloudless sky SIF and cloudless sky GPP are completely different from cloudy sky SIF and cloudy sky GPP and consequently, their relationship may also differ (Miao et al., 2018). Investigating GPP and SIF relationships based only on clear sky data and only on cloudy sky data, without the mix of both, is

justified to better understand their links. Ultimately, not only the weak and statistically significant relationship reported for all biomes on data pooled together across all sites confirmed the PFT-dependent relationships between GPP and $SIF_d$ in this study, but also the significant effects of the year, site and PFT in the relationship between $SIF_d$ and GPP reported in the GLM model further supported this hypothesis. Exploring the newly launched satellite instruments such as OCO-3 and ECOSTRESS and upcoming FLEX and GeoCarb satellite missions which are

planned to have diurnal sampling or fine-spatial resolution (for instance 300 m for FLEX), along with ongoing ground-based and airborne-based SIF and GPP data altogether will improve the abilities to not only better





understand the GPP and SIF relationship but also to completely decouple the effects of driving factors such as leaf morphology and orientation, vegetation physiology, canopy structure and abiotic stress conditions that mediate their relationships at the ecosystem scale.

### 5.2    Synergy between SIF, surface spectral reflectance, and reflectance based-indices for estimating GPP using Random Forest

The second main goal in this manuscript was to explore the synergy between $SIF_d$ from TROPOMI instrument and MODIS surface spectral reflectance and reflectance based-indices namely NDVI, $NIR_v$ and PRI for predicting GPP on data pooled across all sites. For achieving this purpose, four RF regression models were established: RF-R, RF-SIF-R, RF-SIF-R-PFT, and RF-SIF-VI. Except for RF-SIF-R-PFT model, the main advantage of using solely remotely sensed data for estimating GPP is that it can be avoided using information on land cover type and land cover change, as well as meteorological data (J. Xiao et al., 2019).

The current results show that RF-R model (surface spectral reflectance alone) could explain 86% of the variance in tower-based GPP at the daily time scale, whereas RF-SIF-R ($SIF_d$ plus surface spectral reflectance), RF-SIF-R-PFT ($SIF_d$ plus surface spectral reflectance plus PFT), and RF-SIF-VI ($SIF_d$ plus reflectance based-indices) models explain 82%, 83% and 75% of the interannual variabilities in GPP across all sites, respectively. These results suggest that at the seasonal scale spectral reflectance presumably capture the variations in canopy structure, while SIF is highly dependent on variabilities in changes in absorbed photosynthetically active sunlight (APAR). The seasonal variations in canopy structure, especially LAI, are strongly correlated with variations in GPP ((Dechant et al., 2022)). This could justify the strong relationship found between tower-based GPP and the predicted GPP by the RF-R model. On the other hand, SIF is an integrative variable at the seasonal and interannual scales as shown in Figure 9 and on the correlation matrix results (strong contribution of SIF on GPP estimates and high correlation between GPP and SIF compared to each spectral reflectance band). This may explain why SIF, while exhibiting the highest relative importance, fails to improve the GPP estimate. Furthermore, while being limited by its spatial resolution (7 km x 3.5 km), at which SIF may lose its physiological information and most likely reflect phenological, structural and illumination information (Jonard et al., 2020; Kimm et al., 2021), SIF remains a better predictor of GPP than each reflectance band individually. These results also revealed that the RF-SIF-VI have the poorest performance in predicting GPP. This lower performance could be partly due to the well-known saturation of VIs over intense canopies. In addition, the paired t-test did not show any statistically significant difference between RF-R and RF-SIF-R models, which confirms the above hypothesis, which suggests that SIF represents the variations in APAR at these scales. Recently, Pabon-Moreno et al. (2022) used solely Sentinel-2 satellite derived red-edge-based and near-infrared-based vegetation indices and all spectral bands to predict GPP at daily time scale across 54 EC flux sites using a data-driven approach (Random Forest). The authors reported that spectral bands jointly with VIs can inform only 66% of the variance in GPP, which is far less than the here worse performing model (i.e. RF-SIF-VI) in predicting GPP. The daily scale and solely remotely sensed driven RF-R and RF-SIF-R models outperform previous GPP products derived based on data-driven methods (Wolanin et al., 2019; Tramontana et al., 2016; Jung et al., 2019) and process-based model (Jiang & Ryu, 2016; Y. Zhang et al., 2017; Lin et al., 2019), which included even further inputs as predictive variables such as meteorological data, land cover type and land cover change data and were conducted mostly at longer time scales (8-day or monthly time scale) compared to this study. Furthermore, these results are in strong agreement to two recent studies (Cho et al., 2021; X. Li et al., 2021). More specifically, Cho et al. (2021) found that remotely sensed data alone can explain 81% of




GPP variability across four vegetation types, including ENF, EBF, DBF, and MF in South Korea at 8-day time scale and Li et al. (2021) pointed out that instantaneous GPP estimates across 56 flux tower sites could be achieved with a $R^2$ of 0.88 and RMSE of 2.42 µmol $CO_2$ m$^{-2}$ s$^{-1}$ using ECOSTRESS land surface temperature, daily MODIS satellite data and meteorological data from ERAS reanalysis. This study revealed also that GPP prediction can be achieved with high accuracy based on solely remotely sensed data that are widely and publicly available for all.

The RF models could clearly capture the GPP variations at each site encompassing different vegetation types as shown in Figures 6 and 8. Indeed, there are sites, years and vegetation types where tower-based GPP were underestimated, which were the cases for WET and EBF vegetation types. Furthermore, all RF models suffer to estimate accurately tower GPP at *IT-BCi, IT-Cp2* and *SE-Deg* sites, owing most likely to SIF pixel heterogeneities and lower GPP values observed in these sites, along with previous explained issues associated in estimating GPP in crops and tropical stands. Similar results were reported recently in Pabon-Moreno et al. (2022) including eight vegetation types (ENF, CRO, DBF, GRA, WET, MF, SAV, and OSH). The reason behind these poor performances may be also related to difficulties to detect abiotic stress conditions (Bodesheim et al., 2018), underscoring the needs of more research for predicting GPP during extreme-abiotic conditions.

Furthermore, in this study, it is determined what are the main variables contributing to GPP prediction using the four RF models based on the relative importance metric of each model. Yet, it is found that $SIF_d$, the surface spectral reflectance in the NIR band ($B_2$), red band ($B_1$) and far-red band ($B_{13}$), as well as the vegetation type, NDVI and $NIR_v$ seem to provide useful information for the predictions of GPP as shown in Figure 9. $B_2$ and $B_1$ are well-known spectral bands for characterizing vegetation canopy structure, seasonal phenology, canopy scattering and reabsorption due to chlorophyll content within leaves, and consequently have a dominant role in estimating GPP across all sites. The high contribution of $SIF_d$ is presumably due to its integrative role at the seasonal and interannual scales as explained previously (Maguire et al., 2020; Dechant et al., 2022). PRI is known to be implied in the xanthophyll cycle, which is an important photoprotection mechanism and as a driver of GPP (X. Wang et al., 2020; Hmimina et al., 2015; Soudani et al., 2014). However, in this study, the findings evidenced that the contribution of PRI on predicting GPP was weak, which could be explained by the spatial and temporal aggregation of the rapid responses in plant physiological and functional activities, observable at the finer scales (diurnal). Ultimately, the findings in this study suggest that using spectral reflectance bands and SIF for estimating GPP is an important approach for improving GPP predictions compared to GPP products that include meteorological and land cover type information.

### 6.    Conclusion

In this current study, the strength of the linear relationships between tower-based GPP and $SIF_d$ encompassing eight major plant functional types (PFT) at the site and interannual scales was evaluated, and the synergy between $SIF_d$, surface spectral reflectance, and reflectance-based indices namely NDVI, NIRv and PRI to improve GPP estimates using a data-driven modelling approach was examined.

At the site scale, the results showed a strong and statistically significant relationships between $SIF_d$ and GPP (p<0.0001). However, the slopes and intercepts of their relationships were site-dependent, indicating that canopy structure and environmental conditions affect the relationship between GPP and $SIF_d$. The GPP and $SIF_d$ relationship across all sites of the same PFT was considerably significant and was PFT-specific. Furthermore, it



was also found that the relationship between GPP and $SIF_d$ on data pooled across all sites was weak but statistically significant, confirming the PFT dependence of the relationship between $SIF_d$ and GPP. The GLM model results supported this PFT-dependent relationship between GPP and $SIF_d$ as the site, year and PFT have meaningful effects on the slope of the relationship between GPP and $SIF_d$.

This study also demonstrated that the spectral reflectance bands, and $SIF_d$ plus reflectance explained over 80% of the tower-based GPP variance. The RF models were able to represent the GPP seasonal and interannual variabilities across all sites. In addition, from the mean decrease in impurity results obtained from the RF models, it is inferred that the spectral reflectance bands in the near-infrared, red and $SIF_d$ appeared as the most influential and dominant factors determining GPP predictions. In summary, this study provides insights into understanding the strength of

the linear relationships between GPP and SIF across different ICOS flux sites and the use of the daily MODIS surface spectral reflectance and $SIF_d$ TROPOMI on predicting GPP across different vegetation types.

Code and data availability. The computer codes (MATLAB and Python) used in this study are available upon demand from the corresponding author. Observations of carbon fluxes are available through the ICOS Data Portal

services (https://www.icos-cp.eu/data-services p.eu). SIF data from TROPOMI instrument satellite are available through (https://data.caltech.edu/records/1347). Daily MODIS Aqua and Terra spectral reflectance data are available through Google Earth Engine (https://earthengine.google.com/arth Engine). Merged datasets are available on the request of the corresponding author.

Supplement. The supplementary materials related to this manuscript is available as a pdf document.

Author contributions. All authors contributed to the paper conceptualization. HB performed the data collection and preparation. HB and GH performed the data pre-processing, analyses and prepared the figures. HB led the writing of the manuscript with the contributions from all authors. KS, YG and GH supervised the project.

Competing interests. The authors declare that they have no conflict of interest.

Funding. This ongoing Ph.D work is jointly funded by le Centre National d'Études Spatiales (CNES) and ACRI-ST.

Acknowledgements. We thank Philip Köehler and Christian Frankenberg at Caltech for making TROPOMI SIF data available. We would also like to be thankful to all Integrated Carbon Observatory System (ICOS) PIs for providing the site level tower-based GPP data through the ICOS Data Portal services. Site names and locations are listed in Table 1 in Supplementary Material S1. We highly appreciate the supporting funding of CNES and le

Programme National de Télédétection Spatiales (PNTS) across the ECOFLUO and C-FLEX projects, respectively. At last, not the least, we thank EIT Climate-KIC financial supports via the ARISE (Agriculture Resilience, Inclusive, and Sustainable Enterprise) project.

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
