# Peer review of "Synergy between TROPOMI sun-induced chlorophyll fluorescence and MODIS spectral reflectance for understanding the dynamics of gross primary productivity at integrated carbon observatory system (ICOS) ecosystem flux sites"

_EGUsphere, 2022_

## Author Comment (AC1)

Dear Dr. Mukund Palat Rao,

We would like to thank you for taking your time to evaluate our work and foremostly for your interesting and useful comments, questions, remarks and suggestions.

We will try to answer to your questions and we integrated your structural and grammatical suggestions into the manuscript (all answers are in blue color).

Do the RFR style models (RF-F, RF-SIF-R, RF-SIF-R-PFT) outperform the RF-SIF-VI model just because they have more predictor variables (14 vs 4)? Perhaps you can present some analysis on how these models perform on 'validation' data that the models have not been calibrated on?

The RF-R model has 11 inputs, the RF-SIF-R 12 inputs, the RF-SIF-R-FTP 13 inputs and RF-SIF-VI 4 inputs. In the manuscript, we used the adjusted $R^2$, which considers the number of samples and predictor variables in its computation to evaluate and compare the performance of our different models (see line 221, page 6). We also compared the models based on the RMSE, which isn't sensitive to the number of explanatory variables.

We also used out-of-bag predictions of RF to calculate the adjusted $R^2$ and RMSE. We separated our dataset in two datasets: 80% of the data for training and 20% for testing or evaluating the model. Second, on the training dataset we applied a 10-fold cross-validation and 20 iterations to determine the best parameters for each model. Lastly, we evaluated or tested each model on the testing dataset, which were not seen by the model before (see lines 207-213, page 6).

As these points were not clearly stated in the manuscript, the lines 219 to 222 were modified as follow :
*"Ultimately, the linear relationships between SIF and GPP were compared based on the coefficient of determination (R2 ), Root Mean Squared Error (RMSE), and the p-value metrics. The random forest models were evaluated and compared based on out-of-bag adjusted R² and RMSE".*

Page 1, Line 15: "Earliest" studies, based could be "earlier" studies or prior studies?
Line 15: changed to "prior studies"

Page 1, Line 17: "plant functional type" should be plant functional types.
Changed to "plant functional types".

Page 2, Line 14: "which is the amount of flux carbon taken up by vegetation." The word 'amount' is not needed since the work flux assumes an amount.
L 43 :Changed to "which is the carbon flux taken up by vegetation through photosynthesis"

Page 2, Line 53: "Remote sensing is widely used to upscale canopy GPP to landscape, regional, and global scales and at daily scale using reflected sunlight measured by satellite sensors". The "and at daily scale" addition seems a bit awkward. Maybe remote sensing is widely used to upscale daily GPP to landscape….
Changed to: "Remote sensing is widely used to upscale daily GPP to landscape, regional, and global scales using reflected sunlight measured by satellite sensors."

Page 2, Line 60: "and biochemical canopy characteristics (Dechant et al., 2020; PabonMoreno et al., 2022). Although, they suffer." The way the sentence is framed, I think it would flow better as a continuous sentence.
Changed to: "and biochemical canopy characteristics (Dechant et al., 2020; Pabon-Moreno et al., 2022), but they suffer from contamination by atmosphere and saturation in canopy dense ecosystems and are less sensitive to diurnal and daily variations in photosynthetic status resulting from physiological responses induced by rapid changes of abiotic stresses."

Page 3, Lime 83: "Early studies relied on ground-based", should be Earlier studies relying on…
Changed to "Earlier studies relying on"

Page 3, Line 96: "which is on board Sentinel 5-Precursor, represents a novel (???) for understanding:". A missing word after novel? Tool maybe?
Changed to: "which is on board Sentinel 5-Precursor, represents a novel tool for understanding"

Page 3, Line 97: "it provides a quiet high temporal resolution at daily". Quite instead of quiet. However, the word quite is not needed either.
Changed to: "it provides a high temporal resolution at daily scale."

Page 3, Line 104: "comprehensively addressed. Owing to most". This should be once
sentence, or the second sentence should start of as, this is due to the fact or This is
because….
Changed to: "However, to the best of our knowledge, an attempt to study the synergy between those variables have not been comprehensively addressed due to the fact that the relationships between structural and functional components are not linear, and have complex interactions over time and space"

Page 4, Line 8: "data products is given in Supplementary Materials in Tab S2.". the
Authors don't need to do this, but might consider including a column for the spectral band
(i.e. visible (R/G/B), NIR, etc. in the table)..
Here the spectral band column was added in tab S2 as you suggested (see supplementary Material in Tab S2).

| Acronym | Full Name | Wavelengths (nm) | Band name | Spatial Resolution |
|---|---|---|---|---|
| $B_1$ | Surface Reflectance for $B_1$ | 620-670 | Red | 500 m |
| $B_2$ | Surface Reflectance for $B_2$ | 841-876 | NIR | |
| $B_3$ | Surface Reflectance for $B_3$ | 459-479 | Blue | |
| $B_4$ | Surface Reflectance for $B_4$ | 545-565 | Green | |
| $B_5$ | Surface Reflectance for $B_5$ | 1230-1250 | SWNIR | |
| $B_6$ | Surface Reflectance for $B_6$ | 1628-1652 | SWIR | |
| $B_7$ | Surface Reflectance for $B_7$ | 2105-2155 | SWIR | |
| $B_8$ | Surface Reflectance for $B_8$ | 405-420 | Blue | 1 km |
| $B_9$ | Surface Reflectance for $B_9$ | 438-448 | Blue | |
| $B_{10}$ | Surface Reflectance for $B_{10}$ | 483-493 | Blue | |
| $B_{11}$ | Surface Reflectance for $B_{11}$ | 526-536 | Green | |
| $B_{12}$ | Surface Reflectance for $B_{12}$ | 546-556 | Green | |
| $B_{13}$ | Surface Reflectance for $B_{13}$ | 662-672 | Red | |
| $B_{14}$ | Surface Reflectance for $B_{14}$ | 673-683 | VNIR | |
| $B_{15}$ | Surface Reflectance for $B_{15}$ | 743-753 | VNIR | |

| $B_{16}$ | Surface Reflectance for $B_{16}$ | 862-877 | NIR |
| --- | --- | --- | --- |

Page 6, Line 230-232: "Overall, SIFd was significantly related with tower-based GPP at the
site level and at the daily timescale (as p<0.0001 was statistically highly significant),
except for IT-Cp2 site of which GPP and SIFd relationship was insignificant and weak (R2= 0.001, p≤0.60)". This is of course quite subjective, but despite some of the sites being statistically significant I would not call these relationships as being strong. The reason for the statistical significance and p-value is being driven by the high sample size. In particularly, I would add GF-Guy to the list of sites where there is no relationship between GPP-SIFd. The correlation needed to get an R^2 of 0.2 is around 0.15 which is still quite weak. I would also then add FR-Mej, FR-EM2, and FI-Var to the list of sites with a weak
relationship. I know this is mentioned a bit later, but maybe an easier way to frame it
would be to not mention the weak relationship at IT-CP2 in the beginning, but then
mention all these sites together at the end of the paragraph?

*This part was reframed as follows (lines 230-236): "Overall, SIFd was significantly related with tower-based GPP at the site level and at the daily timescale (p<0.0001). However, Figure 2 indicates that the slopes and intercepts of the linear regression between GPP and SIFd are site-dependent, suggesting that the difference in plant functional types and spatial heterogeneity across sites may significantly affect the relationships between GPP and SIFd. The strongest relationships were found at DK-Sor and FR-Fon, which are DBF vegetation type sites, with R2 values of 0.81 (p<0.0001) and 0.66 (p<0.0001). The weakest linear relationships were recorded at FI-Var, FR-EM2 and FR-Mej sites, and no significant relationship was found at GF-Guy and IT-Cp2.".*

Page 7, section 4.1.2 I like the progress from the site level (Section 4.1.1) to PFT level SIFd-GPP relationships. However, the way the PFT level relationships are presented, don't seem to actually allow us to closely examine within PFT spread in the R2's, slopes, and intercept. For example, in Table 2, all sites of a PFT are lumped together. If the authors with to highlight the within PFT spread, once option could be to include boxplots by PFT for the R2, slope, and intercept for the SIFd-GPP relationships.

We did not include the boxplot of $R^2$, slopes and intercepts of the GPP vs $SIF_d$ for sites from the same PFT, because, we have low number of sites for some PFT, including OSH (1 site), MF (2 sites), EBF (2 sites), and GRA (3 sites). In addition, the $R^2$, slopes and intercepts of the relationships between GPP and $SIF_d$ for each PFT and site are detailed in Tab S3.

Page 9, Section 4.5, Line 284, Supplementary Fig. S1: I would recommend changing the
figure to have a diverging colour-bar. The gradient colour-bar from ~-0.65 to 1 is not
intuitive to me and hard to visualize.

In the figure below, the gradient colour-bar for the correlation matrix was changed, as you recommended. The figure was added to the Supplementary Material draft.

[Figure]

Page 12, Table 3: would benefit from a vertical line separating RF-R, RF-SIF-R, and N, and other similar vertical line in the lower panel
The vertical lines separating RF-R, RF-SIF-R and N were added in table 3.

| PFT | Sites | N | RF-R | | | | RF-SIF-R | | | |
|-----|-------|---|------|------|-----------|------|----------|------|-----------|------|
| | | | Adj. $R^2$ | Slope | Intercept | RMSE | Adj. $R^2$ | Slope | Intercept | RMSE |
| CRO | 9 | 1171 | 0.78 | 1.03±0.03 | 0.00±0.24 | 2.67 | 0.75 | 1.01±0.03 | 0.08±0.26 | 2.89 |
| DBF | 6 | 748 | 0.92 | 1.02±0.02 | -0.23±0.18 | 1.41 | 0.90 | 1.05±0.02 | -0.52±0.21 | 1.61 |
| EBF | 2 | 188 | 0.77 | 0.93±0.07 | 1.01±0.83 | 1.23 | 0.68 | 0.90±0.09 | 1.58±0.99 | 1.45 |
| ENF | 13 | 1385 | 0.85 | 1.01±0.02 | -0.01±15 | 1.29 | 0.78 | 1.06±0.03 | -0.23±0.19 | 1.54 |
| GRA | 3 | 364 | 0.81 | 1.02±0.05 | -0.02±32 | 1.64 | 0.76 | 1.07±0.06 | -0.17±0.38 | 1.87 |
| MF | 2 | 117 | 0.84 | 1.05±0.08 | -0.15±0.76 | 1.49 | 0.82 | 1.12±0.10 | -0.62±0.83 | 1.56 |
| OSH | 1 | 317 | 0.91 | 1.02±0.04 | -0.09±0.22 | 0.99 | 0.88 | 1.01±0.04 | 0.01±0.24 | 1.10 |
| WET | 4 | 599 | 0.92 | 0.98±0.02 | -0.15±0.10 | 0.85 | 0.84 | 0.98±0.03 | -0.37±0.15 | 1.17 |
| ALL | 40 | 4889 | 0.86 | 1.02±0.01 | -0.09±0.08 | 1.72 | 0.82 | 1.04±0.01 | -0.19±0.10 | 1.94 |
| | | | RF-SIF-VI | | | | RF-SIF-R-PFT | | | |

| PFT | Sites | N | Adj. R$^2$ | Slope | Intercept | RMSE | Adj. R$^2$ | Slope | Intercept | RMSE |
|-----|-------|---|-----------|-------|-----------|------|-----------|-------|-----------|------|
| CRO | 9 | 1171 | 0.70 | 1.03±0.04 | 0.01±0.29 | 3.14 | 0.75 | 1.00±0.03 | 0.12±0.26 | 2.87 |
| DBF | 6 | 748 | 0.84 | 1.05±0.03 | -0.58±0.28 | 2.06 | 0.91 | 1.04±0.02 | -0.40±0.21 | 1.56 |
| EBF | 2 | 188 | 0.51 | 0.77±0.11 | 3.42±1.14 | 1.80 | 0.72 | 0.96±0.09 | 0.74±0.98 | 1.37 |
| ENF | 13 | 1385 | 0.66 | 1.02±0.04 | 0.10±0.24 | 1.92 | 0.79 | 1.08±0.03 | -0.39±0.19 | 1.5 |
| GRA | 3 | 364 | 0.69 | 0.98±0.07 | 0.02±0.43 | 2.11 | 0.77 | 1.07±0.06 | -0.29±0.38 | 1.84 |
| MF | 2 | 117 | 0.71 | 1.04±0.12 | 0.04±1.07 | 2.00 | 0.82 | 1.12±0.09 | -0.73±0.84 | 1.56 |
| OSH | 1 | 317 | 0.83 | 0.98±0.05 | 0.21±0.29 | 1.33 | 0.89 | 1.02±0.04 | -0.06±0.24 | 1.08 |
| WET | 4 | 599 | 0.72 | 0.88±0.04 | -0.39±0.21 | 1.54 | 0.88 | 1.05±0.03 | -0.29±0.12 | 0.99 |
| ALL | 40 | 4889 | 0.75 | 1.03±0.02 | -0.18±0.12 | 2.28 | 0.83 | 1.03±0.01 | -0.15±0.09 | 1.89 |

Page 15, Line 431: "it can be avoided", maybe better phrased as "we don't need to rely on land cover type….and meteorological data"?
Changed to: "we do not need to rely on land cover type and land cover change, and meteorological data."

Page. 16, Line 465: ERA5 instead of ERAS?
Changed to: "from ERA5 reanalysis."

---

## Author Comment (AC2)

Dear reviewer,

We would like to thank you for taking your time to evaluate our work and foremostly for your interesting and useful comments, questions, remarks and suggestions.

We will try to answer your questions and we will integrate your suggestions into the manuscript (all answers are in blue color).

The authors claim there is a linear correlation between SIFd and GPP both in the site and the PFT levels. However, a quick look at the figures (2-4) shows that in most cases at some point the SIFd-GPP relation reaches saturation. The authors did not mention this even once in their manuscript. Several works are demonstrating this relation and discuss its meaning (see He et al., 2020 for example), however, the authors here ignore it and refer to it as a linear relation. Moreover, in many cities and PFT, the linear correlation is also low for the same reason.

While non-linear relationships have been shown at canopy scale (He et al. 2020, Kim et al. 2021), when using satellite data, the added noise and inherent linearization at larger scales (3.5*7.5 km in our case) makes it hard to fit non-linear model across a diverse set of sites. In our case, non-linear models do not show a clear improvement in performance, as shown in the following figure and table :

[Figure]

| | linear regression | | Hyperbolic model (Kim et al. 2021) | | linear+square root model (He et al. 2020) | |
|---|---|---|---|---|---|---|
| Site name | R² | RMSE | R² | RMSE | R² | RMSE |
| BE-Bra | 0.592 | 2.00 | 0.511 | 2.19 | 0.590 | 1.94 |
| BE-Lcr | 0.204 | 4.06 | 0.198 | 4.09 | 0.211 | 4.04 |
| BE-Lon | 0.274 | 6.05 | 0.250 | 6.16 | 0.263 | 6.14 |
| BE-Vie | 0.202 | 2.44 | -0.160 | 2.96 | 0.213 | 2.41 |
| CH-Dav | 0.153 | 1.74 | -0.597 | 2.40 | 0.163 | 1.75 |
| CZ-BK1 | 0.354 | 3.38 | -0.345 | 4.91 | 0.232 | 3.53 |
| CZ-Lnz | 0.416 | 3.18 | 0.169 | 3.80 | 0.346 | 3.22 |
| CZ-Wet | 0.196 | 3.68 | 0.038 | 4.04 | 0.219 | 3.64 |
| DE-Geb | 0.526 | 3.80 | 0.546 | 3.73 | 0.570 | 3.66 |
| DE-Gri | 0.278 | 3.26 | 0.222 | 3.39 | 0.255 | 3.31 |
| DE-Hai | 0.569 | 2.91 | 0.579 | 2.88 | 0.544 | 2.96 |
| DE-HoH | 0.346 | 4.14 | 0.261 | 4.40 | 0.306 | 4.27 |
| DE-Hzd | 0.415 | 2.73 | 0.317 | 2.96 | 0.379 | 2.82 |
| DE-Kli | 0.310 | 3.85 | 0.293 | 3.90 | 0.314 | 3.88 |

| | | | | | | |
|---|---|---|---|---|---|---|
| DE-RuR | 0.174 | 3.80 | 0.024 | 4.14 | 0.169 | 3.86 |
| DE-RuS | 0.289 | 6.22 | 0.296 | 6.20 | 0.272 | 6.34 |
| DE-RuW | 0.159 | 2.61 | -0.959 | 4.00 | 0.000 | 2.76 |
| DE-Tha | 0.558 | 1.61 | 0.178 | 2.20 | 0.495 | 1.68 |
| DK-Sor | 0.808 | 2.73 | 0.787 | 2.89 | 0.800 | 2.74 |
| FI-Hyy | 0.274 | 2.68 | -0.111 | 3.33 | 0.304 | 2.54 |
| FI-Sii | 0.318 | 1.10 | 0.082 | 1.28 | 0.296 | 1.09 |
| FI-Var | 0.130 | 2.00 | -1.146 | 3.14 | -0.008 | 2.08 |
| FR-Aur | 0.225 | 3.65 | 0.221 | 3.66 | 0.198 | 3.76 |
| FR-Bil | 0.169 | 3.13 | -0.160 | 3.70 | 0.114 | 3.28 |
| FR-EM2 | 0.111 | 4.85 | 0.058 | 5.00 | 0.089 | 4.94 |
| FR-Fon | 0.664 | 2.71 | 0.645 | 2.79 | 0.653 | 2.72 |
| FR-Hes | 0.499 | 3.78 | 0.495 | 3.80 | 0.489 | 3.81 |
| FR-LGt | 0.428 | 2.29 | 0.422 | 2.30 | 0.424 | 2.27 |
| FR-Mej | 0.045 | 5.22 | 0.060 | 5.18 | 0.066 | 5.17 |
| GF-Guy | 0.020 | 2.63 | 0.010 | 2.65 | -0.045 | 2.72 |
| IT-BCi | 0.158 | 6.11 | 0.001 | 6.67 | 0.131 | 6.16 |
| IT-Cp2 | 0.001 | 2.01 | -4.463 | 4.73 | -0.397 | 2.40 |
| IT-Lsn | 0.538 | 2.10 | 0.488 | 2.22 | 0.532 | 2.09 |
| IT-SR2 | 0.206 | 2.95 | -0.991 | 4.68 | 0.201 | 2.96 |
| IT-Tor | 0.424 | 2.83 | 0.345 | 3.02 | 0.383 | 2.90 |
| NL-Loo | 0.237 | 2.19 | -0.349 | 2.93 | 0.352 | 1.96 |
| SE-Deg | 0.353 | 0.54 | 0.025 | 0.66 | 0.251 | 0.54 |
| SE-Htm | 0.291 | 2.72 | -0.114 | 3.41 | 0.227 | 2.86 |
| SE-Nor | 0.508 | 2.14 | 0.308 | 2.54 | 0.522 | 2.09 |
| SE-Svb | 0.198 | 2.75 | -0.542 | 3.81 | 0.080 | 2.68 |

While the two tested non-linear models do not show significantly lower RMSE overall, they exhibit a strong instability, and cannot be accurately fitted on all sites. As our goal is to compare relationships between sites and considering the level of noise in the TROPOMI SIF data, we see no clear benefit in using a non-linear model which only brings in marginal improvement over a few sites at the expense of a loss in genericity.

I'm not sure what is the added value of the pooled graph of all PFTs vs. SIFd (Fig. 4)

Within this figure, we would like to evaluate the genericity of the relationship between GPP and SIFd across the study sites, which is demonstrated by the low $R^2$ value found on data pooled across all sites.

It is very hard to estimate the performance of the different models vs. EC GPP in Figure 8. Please consider reducing the size of the dots and making them transparent.

The size of the dots has been changed and we made them transparent.

[Figure]

Line 399: Please try to explain why the high correlation in the DBF and OSH PFT's

We briefly tried to explain this high correlation in line 399. The main explanation is that in DBF and OSH (one sample of vineyard plantation) biomes, there are explicitly marked seasonal and phenological changes compared to EBF or ENF forest where there is greenness all time. Thus, in DBF and OSH biomes SIF signal may easily capture the variations in LAI and APAR and consequently display a high correlation between GPP and SIF$_d$ (added to: line 405-408).

Line 410: This is not clear to me, the authors mention in the methods section that they took out the cloudy day data. Line 414: So, why not do that in your data?

The representativity of satellite SIF data which needs to be filtered for cloud coverage is indeed a limitation of the current study. This limitation is inherent to the use of satellite data, and can only be lifted through the collection of ground-based SIF data across diverse ecosystems.

Line 436: it is problematic to say there is a difference in the models while earlier you mentioned there was no statistical difference between them (line 296). Line 447: same comment as above.

The line 296 was indeed unclear. We edited it as follows:

"A paired t-test realized between the four models based on the adjusted $R^2$ performance revealed that the difference in mean **adjusted $R^2$** between RF-R and RF-SIF-R, RF-R and RF-SIF-R-PFT, and RF-SIF-R and RF-SIF-R-PFT models was not statistically significant. In other words, these three FR models have the same performance."

We have reframed the line 436 as follows to clarify it:

"The current results show that the RF-R (surface spectral reflectance alone), RF-SIF-R (SIFd plus surface spectral reflectance) and RF-SIF-R-PFT (SIFd plus surface spectral reflectance plus PFT) models explain a non-significative different percentage of the variance in tower-based GPP at the daily time scale (82~86%), whereas the RF-SIF-VI (SIFd plus reflectance based-indices) explains 75% of the interannual variabilities in GPP across all sites."

As for line 447, it is based on a difference in relative importance rather than a difference in $R^2$ (see Figure 9). Unlike the differences in $R^2$ which aren't statistically different and cannot be interpreted, the relative importance depends on input variables and can be interpreted. This sentence was edited as follows to clarify it:

"SIF remains a better predictor of GPP than each reflectance band individually (Fig. 9)."

---

## Author Response (AR2)

Responses to comments: our replies are all in magenta color.

Dear,
We would like to thank you for taking your time again to assess our work and more importantly for your useful comments, remarks and suggestions. We tried to answer to your questions.

**Responses to Reviewer 1:**

The revised version of the manuscript did have some improvement in some aspects. However, the authors' two main problems remain without an answer: the first one is the forcing of linear relation between the SIFd and GPP data. As the data demonstrated, in most of the sites at the high SIFd values (>~0.5) the GPP is already saturated. Referring it as a linear relation (even though what the authors actually have is a correlation…), is problematic. A simple plot of the observed data residuals over the fitted line will demonstrate it. The authors have many ways to solve this issue: normalize the values to get linear correlation, fit another equation to the data or even simply try to explain why they prefer to describe it as a linear regression, but they did not do it. The title of the work is talking about "…understanding the dynamics of gross primary productivity…", however the work doesn't try to understand this dynamic. In my opinion, analysing the sites or PFT real SIF-GPP dynamic (not always linear) can improve this paper.

This was added to page 6, line 185-192: "We used a hyperbolic model to relate GPP to $SIF_d$ following Damm et al. (2015). This hyperbolic model approximates only the data behaviour and supports the theoretical argument that GPP saturates at moderate and high $SIF_d$ level: $GPP = a \times \frac{SIF_d}{SIF_d + b}$ ; where a and b are fitted parameters. It is worth noting that a linear model between GPP and $SIF_d$ was also investigated, and the results are provided in supplementary materials. Before relating GPP to $SIF_d$ using this hyperbolic model at each site, SIF values equal or less than zero were discarded. Afterward, the same model was fitted on PFT scale by pooling all data across all sites for the same PFT."

This was added to page 7, line 231-232: "evaluate the strength of the relationships between tower-based GPP and $SIF_d$ encompassing different vegetation types at site level. To do so, a hyperbolic model was used to relate GPP to $SIF_d$ at each site."

This was added to page 7, line 233-234: "Overall, the results revealed a hyperbolic relationship with relatively saturating GPP in presence of moderate to high $SIF_d$."

This was added to page 7, line 236-240: "The strongest relationships were found at DK-Sor, FR-Fon, DE-Tha, SE-Nor and BE-Bra, which are DBF, ENF and MF vegetation type sites, with $R^2$ values being between 0.64 and 0.87 (p<0.0001). The weakest relationships were recorded at FI-Var, FR-EM2 and DE-RuW sites, and no significant relationship was found at GF-Guy, IT-Cp2 and FR-Mej."

This was added to page 7, line 242-245: "Note that the independent assessment considering the linear model to relate $SIF_d$ to GPP at each site, and each PFT and on data pooled across all sites revealed a relatively consistent lower goodness of fit, justifying the use of a hyperbolic model (see Supplementary Material Tab S4 and S5, Figures S1, S2 and S3)."

This was added to page 8, line 251: "The black dotted line represents the hyperbolic fit between GPP and $SIF_d$."

This was added to page 9, line 252-254: "the hyperbolic relationship between GPP and $SIF_d$ was strongest for OSH, DBF and MF, with $R^2$ of 0.61, 0.59 and 0.52, respectively, and the lowest for EBF with $R^2$ of 0.06. This result suggests that the relationships between GPP and $SIF_d$…"

This was added to page 10, line 275-277: "These findings support that the GPP and $SIF_d$ relationship is considerably influenced by the site PFT and the interannual variations in $SIF_d$."

This was added to page 15, line 372-373: "However, the hyperbolic fit between tower-based GPP and $SIF_d$ vary significantly across sites, which suggests a site-specific relationship."

This was added to page 15, line 387-394: "Furthermore, these results are also in good agreement with several studies carried out with instantaneous ground-based measurements at different vegetation types, sites and locations (Kim et al., 2021, Damm et al., 2015; He et al., 2020, Gu, Han, et al., 2019). For instance, Kim et al. (2021) pointed out that a hyperbolic model could explained better the relationships between GPP and SIF in an evergreen needle forest and Damm et al. (2015) showed similar results in croplands, mixed temperate forests and grassland vegetation types. One of the most plausible explanations is that GPP might reach saturation at high light, while SIF tends to keep increasing with PAR. It is also paramount to mention that the saturation of optical signal is a common issue in remote sensing, which can explain part of the lower relationships found in the EBF sites."

This was added to page 15, line 396-399: "Yet, the hyperbolic relationships between GPP and $SIF_d$ vary considerably across PFT, suggesting a PFT-specific relationship. The relationship between GPP and $SIF_d$ is driven by the ratio of canopy photosynthesis and fluorescence yield, along with the canopy escape probability fraction of SIF photons from canopy to sensor "

……………………………………………………………………………………………………………
……………………………………………………………………………………………………………

**Responses to reviewer 2:**

Dear Dr. Mukund Palat Rao,
We would like to thank you for taking your time again to evaluate our work and foremostly for your interesting comments, remarks and suggestions so far.
We integrated your structural and grammatical suggestions into the manuscript (all answers and changes are in magenta color).

Page 6, Line 190 & Page 9 Line 267 Generalizability and not genericity

Page 6, line 190 & page 9 Line 267 'genericity' has changed to "generalizability"

Tab S3, the caption "NIR denotes for near-infrared, SWNIR for shortwave near-infrared, SWIR shortwave infrared, and VNIR visible near-infrared." Is not needed for this table. It is however needed for Table S2 instead.

Tab S3, caption "NIR denotes for near-infrared, SWNIR for shortwave near-infrared, SWIR shortwave infrared, and VNIR visible near-infrared." Has transferred to Tab S2.

NB: It is worth noting that the order number of the figures and Tables has changed in the supplementary material and all changes are underlined in magenta color.

Some others minor modifications were made by rereading the MS:

This was added to page 1, line 27-29: "The synergy between $SIF_d$ and MODIS based reflectance (R) and VIs to improve GPP estimates using a data-driven modelling approach was also evaluated."

This was added to page 1, line 32-33: "Using Random Forest Regression models (RF) with GPP as output and the aforementioned variables as predictors (R, SIFd and VIs),…"

This was added to page 1, line 36: "the relative variable importance of predictors of GPP…"

This was added to page 4, line 134: "located in French Guiana."

This was added to page 11, line 315: "even if the $R^2$ remain statistically significant at 5% probability level"

*In page 15, line 368* "**Strength of the linear relationship between GPP and SIF$_d$ at site level and PFT scale**" was changed to "**Strength of the relationship between GPP and SIF$_d$ at site and PFT levels**"

In page 16, line 434 "**Synergy between SIF, surface spectral reflectance, and reflectance based-indices for estimating GPP using Random Forest**" was changed to "**Synergy between SIF$_d$, R and VIs for estimating GPP using Random Forest**"

---

## Author Response (AR3)

Responses to comments: our replies are all in magenta color

Dear Dr. Eyal Rotenberg as editor and fellow reviewers,

We would like to thank you for not only taking your valuable time to asses our MS, but also for your useful and practical scientific added-value discussions, comments, remarks and suggestions, that we had even through direct emails. We tried to answer your comments.

Responses to editor:

Dear Hamadou and the co Authors,

After lengthy discussions, sometimes even via direct exchange emails, I am happy to let you know that the paper, from my side, is almost ready for publication. I think this process helps much to strengthen and clarify the article, which you and the readers will gain. Left are the following two comments which you are asked to consider:

1. In your answer to reviewer 1, referring to lines 185 -192, I found the below part of the sentence unclear and needs your clarification:

"This hyperbolic model approximates only the data behaviour and supports the theoretical…".

2. Challenging is to read Figure 6 legends; consider using common legend, for the X-axis and the Y-axis, instead of the current format.

I sincerely hope that the paper will be well received by the scientific community, as the reviewers received it, and I wish you the best in your future activities.

Kind regards, Eyal.

This was changed to page 6, line 185-192: " We used a hyperbolic model to relate GPP to $SIF_d$ following (Damm et al., 2015; Kim et al., 2021)". The phrase was deleted because it described the results, instead of explaining the methodology used.

Challenging is to read Figure 6 legends: "Considering common legend, for X and Y axes, the Figure 6 was changed".

Note that for the same raison mentioned in Figure 6, all figures regarding observed GPP vs RF predicted GPP was changed by keeping common legend for X and Y axes within the MS and Supplementary Materials as well.